# The Microbial Composition of Bovine Colostrum as Influenced by Antibiotic Treatment

**DOI:** 10.3390/antibiotics14121217

**Published:** 2025-12-03

**Authors:** Ruth Conboy-Stephenson, Dhrati Patangia, Kevin Linehan, R. Paul Ross, Catherine Stanton

**Affiliations:** 1Teagasc Food Research Centre, Moorepark, Fermoy, P61 C996 Co. Cork, Ireland; 2School of Microbiology, University College Cork, T12 Y337 Cork, Ireland; 3APC Microbiome Institute, University College Cork, T12 YT20 Cork, Ireland

**Keywords:** bovine colostrum, dry cow therapy, microbiota, antibiotic resistance, parity, farm

## Abstract

**Background/Objectives**: Bovine colostrum, the initial milk produced by cows postpartum, contains an array of key nutritional, immune, and microbial components that support the calf’s physiological development, immune maturation, and intestinal colonization. The composition and quality of colostrum can be influenced by multiple factors, including seasonal variation, breed, parity, and farm management practices. This study investigated the microbial profile of Irish bovine colostrum and the influence of antibiotic therapy and parity. **Methods**: Bovine colostrum samples were collected from five Irish dairy farms that implemented different methods of dry cow therapy (DCT): natural or blanket. For blanket DCT, four of the five farms administered intramammary antibiotics at the start of the drying off period. Two farms administered a fourth-generation cephalosporin, cefquinome, and two farms used an antibiotic of the penicillin class, with the active ingredients consisting of procaine benzylpenicillin, penethamate hydriodide, and framycetin sulphate. One farm did not administer antibiotics but applied a teat sealant (natural DCT). Following calving, colostrum samples from 90 healthy dairy cows were analysed. **Results**: 16S rRNA sequence analysis revealed Firmicutes, Actinobacteriota, Bacteroidota, and Proteobacteria as the most abundant phyla across all treatment groups, with *Acinetobacter*, *Corynebacterium*, *Facklamia*, *Jeotgalicoccus*, *Lactococcus*, *Leuconostoc*, *Psychrobacter*, and *Staphylococcus* dominating at genus level. Parity did not significantly affect the microbial composition in this study, but antibiotic treatment did. Cows receiving no antibiotics showed distinct microbial clustering compared with antibiotic-treated cows (β-diversity, *p* < 0.001). Microbial diversity also differed between the antibiotic-treated groups, with significant changes in both α-diversity (*p* < 0.01) and β-diversity (*p* < 0.001), suggesting that the choice of antibiotic may also influence the microbiota. An influence of farm was also observed. Differential abundance analysis showed no increase in mastitis-associated genera in colostrum following natural DCT, although increased abundance was demonstrated with blanket DCT. **Conclusions**: Our findings substantiate the diverse and unique microbial composition of bovine colostrum. The data indicate that the microbial profile of bovine colostrum is influenced by antibiotic treatment administered during the dry period and affirms the latest policies inhibiting prophylactic antibiotic administration. Future studies should elucidate strain level changes in the colostrum microbiota following on-farm antibiotic use, assess the associated risks of antimicrobial resistance, and explore non-antibiotic alternatives for drying off cows. Evidently, the microbial composition of bovine colostrum is influenced by farm management strategies and optimizing these measures may further increase the valuable constituents of bovine colostrum and confer added health benefits to the new-born calf.

## 1. Introduction

Colostrum is the initial milk produced by mammals postpartum. Bovine colostrum is a complete source of nutrition for the new-born calf, meeting full nutritional and energy requirements. Colostrum is rich in macro- and micro-nutrients, in addition to a range of bioactive compounds and a unique community of microorganisms [1,2]. Bovine colostrum is compositionally unique to mature milk, with elevated levels of protein (15% vs. 3%), fat (6–7 vs. 3–4%), and lower carbohydrate levels (2.5%) [3]. In addition, a range of essential minerals and vitamins are higher in colostrum [1], including calcium, phosphorus, sodium, chloride, and magnesium [3,4,5], whilst concentrations of potassium and manganese are lower [4,5]. Similarly, a greater concentration of vitamins B2, B12, A, E, and D have been observed in colostrum [3,5]. Importantly, the immunoglobulin (Ig) content is markedly richer, comprising 70–80% of colostrum’s total protein content [1,6]. IgG is the most abundant class, accounting for 80–90% of the total Ig content [3,7]. In addition, colostrum is known for its high availability of bioactive compounds including lactoferrin, lysozyme, and lactoperoxidase [1,3,4,5,8]. These compounds have protective functions for the calf, due to their bacteriostatic and germicidal activity [8]. Patently, colostrum provides increased nutritional value. Bovine colostrum facilitates basic cell function, thermoregulation, maturation of the immune response, and gastrointestinal and muscle development [4,9]. Colostrum consumption in calves is associated with reduced incidences of infection, diarrhoea, respiratory disorder, and mortality [10,11], and increased immune protection, growth performance, and milk yield [10,11,12]. Indeed, the new-born calf is reliant on colostrum to acquire immune protection through passive transfer of immunoglobulins [13,14,15]. Thus, receipt of colostrum is essential for the development and survival of the new-born calf.

In addition to being nutrient dense, colostrum is also host to a rich microbial population, which is passed to the new-born upon ingestion, initiating gut colonization [5,16]. Intestinal colonization by colostrum consumption increased levels of *Bifidobacterium* in the colon and reduced levels of the pathogenic microbes *Escherichia coli* and *Escherichia-shigella*, highlighting its role in immune defence [17]. Although still under investigation, the origin of the bovine colostrum microbiota has mainly been linked with the udder skin and teat canal, and the surrounding farm environment [18,19,20]. In addition, the existence of an entero-mammary pathway has been suggested; however, this is yet to be established [20,21,22,23]. To date, studies are largely in agreement that colostrum is dominated by the same four phyla; Firmicutes, Bacteroidetes, Proteobacteria, and Actinobacteria [24]. However, at lower taxonomic levels, greater discordance exists. A recent review identifies *Acinetobacter*, *Pseudomonas*, and *Staphylococcus* as the most commonly identified bacterial genera in colostrum [4]. Even greater variability exists at the species level. Recent developments in sequencing technology have enabled a more comprehensive insight into the microbial population associated with the bovine mammary gland. Indeed, shotgun sequencing has confirmed the presence of microbes at the species level in bovine colostrum, with both pathogenic and probiotic potential [25]. These include bacteria with beneficial properties widely used as starter cultures in the food industry, such as *Leuconostoc mesenteroides*, *Lactococcus lactis*, and *Lacticaseibacillus paracasei*, and known pathogenic strains, *Pseudomonas aeruginosa*, *E. coli,* and *Listeria monocytogenes* [26,27,28,29]. The bacterial content of colostrum is often screened as an indicator of microbial quality and infection status. In colostrum, a total plate count (TPC) and total coliform count (TCC) of <100,000 CFU mL^−1^ and <10,000 CFU mL^−1^, respectively, are the acceptable thresholds [4]. In bulk tank milk, a somatic cell count (SCC) > 400,000 cells mL^−1^ exceeds the limit for milk intended for human consumption according to European Union (EU) legislation [30]. It is widely accepted that a SCC > 200,000 cells mL^−1^ at herd level is indicative of intermammary infection [31]. Although current legislation refers to bovine milk, SCC is also an effective way of determining the quality of colostrum [32].

The nutritional and bioactive composition of colostrum not only facilitates adequate growth and development in the offspring, but also provides immune protection [8,33]. Van Hese et al. suggests 4 L of high quality colostrum for new-born calves within 6 h of parturition will confer sufficient serum IgG levels (10 g/L) to ensure passive immunity [16]. When the offspring does not receive adequate colostrum of high quality, failed transfer of Igs can occur, decreasing the chance of survival. In addition, high quality colostrum influences intestinal colonization of the neonate, promoting the growth of beneficial bacteria [5]. Clearly, high quality colostrum in the initial critical period ensures successful passive transfer and results in improved health of the calf. However, in the United States and Brazil, only 39% and 22.6% of bovine colostrum samples, respectively, met the industry recommendations for both IgG and microbial growth [34,35]. Furthermore, in Northern Ireland, 44% of colostrum samples had <50 mg mL^−1^ IgG and 81% had a TPC > 100,000 cfu mL^−1^ [13]. Evidently, further optimization of farm management strategies is needed to ensure the production of bovine colostrum with satisfactory microbial and nutritional value.

A multitude of factors can influence the composition and quality of bovine colostrum, including seasonal variation [6,36,37], maternal nutrition [13,38], breed [39,40], and herd size [41]. In addition, parity is known to influence the nutritional and immunological quality of colostrum [6,13,36,37,39,40,42,43]. However, evidence regarding its impact on the colostrum microbiota remains inconsistent [16,44,45]. Furthermore, although some influential factors are predetermined, others can be modified with farm management practices [37], in particular, DCT [37,42]. The dry period is a non-lactating period, referring to the time between the last lactation pre-calving and the first lactation post-calving. This interval is essential for the optimization of mammary gland health and milk production, facilitating the regeneration and repair of mammary epithelial cells [44]. DCT frequently includes the administration of intramammary antibiotics. In addition, an intramammary teat sealant is often used in combination with antibiotics or independently (natural DCT), mimicking the naturally formed keratin plug in ruminants. Teat sealant is usually composed of bismuth subnitrate and forms a physical barrier to protect against invading pathogens [46]. There are two established protocols to antibiotic DCT—selective and blanket. Selective DCT limits antibiotic administration to cows with an ongoing infection or at imminent risk of infection during the dry period. Blanket DCT refers to the prophylactic administration of antibiotics, regardless of the cow’s infection status. However, as of 2020, an EU directive has prohibited prophylactic antibiotic administration on farms due to growing concerns regarding the rise in antibiotic resistance [47]. Conversely, blanket DCT is still practiced in other parts of the world [48,49,50]. The primary infection driving the need for antibiotic administration is bovine mastitis, which is the inflammation of the mammary parenchyma due to bacterial infection. Mastitis is a known economic burden to the dairy sector and an ongoing challenge for farmers and herd health [51]. The main mastitis-causing pathogens include *Corynebacterium bovis*, *Staphylococcus aureus*, *Streptococcus uberis*, *Streptococcus dysgalactiae,* and *Streptococcus agalactiae* [46,52]. The use of antibiotics when treating intramammary infection is considered a key contributor to the growing emergence and spread of antimicrobial resistance. Indeed, antibiotic-resistant bacteria have previously been isolated from bovine milk [53,54]. Moreover, bovine colostrum and milk are suggested reservoirs for antibiotic-resistant genes [25,55,56]. The potential vertical transfer of antimicrobial resistance to the environment and food-chain makes antibiotic administration on dairy farms a One Health issue.

In Ireland, dairy farming contributes EUR 16 billion to the Irish economy and is responsible for 85,000 jobs [57,58]. In addition, the number of dairy cows in Ireland increased by 34.8% in 2020 since 2013, accounting for over 1.5 million dairy cows [59]. As a global leader in the dairy industry, it is of vital importance to not only protect but also optimize the farm and dairy industry through the prioritisation of animal health and milk quality. A greater understanding of the farm management practices that influence the microbial quality of bovine colostrum will enable improved on-site management practices and herd health. In this study, we investigated the microbial composition of bovine colostrum from healthy cohorts of Irish dairy cows. Our primary objective was to compare the effect of blanket and natural DCT on the microbial profile of bovine colostrum. We also determined the effect of parity and individual farm variation.

## 2. Results

In this study, microbial DNA was extracted from bovine colostrum samples. Samples were grouped according to antibiotic treatment. 16S rRNA sequencing was performed to determine the effect of DCT (blanket vs. natural) on the microbial profile of colostrum.

### 2.1. Microbial Composition of Bovine Colostrum Following Dry Cow Therapy

Relative abundance analysis revealed a total of 12 phyla in bovine colostrum. Across all three treatment groups (CEF, NOAB and UBRORED), the most prevalent phyla were identified as Firmicutes (48%), Actinobacteriota (22%), Bacteroidota (7%), Proteobacteria (21%), and Patescibacteria (1%) (see Figure 1). The remaining phyla were identified at abundances of <0.5% and considered to be of low abundance and grouped as “Other”.

Pairwise comparisons of the distribution of relative abundance at phylum level between DCT treatment groups revealed significant differences in the abundance of Firmicutes, Actinobacteriota, and Patescibacteria (see Figure 2). A lower abundance of Firmicutes was observed in the NOAB group when compared with both the CEF and UBRORED groups (*p* < 0.05). The NOAB group had a higher abundance of Bacteroidota than the CEF (*p* < 0.001) and UBRORED (*p* < 0.05)-treated groups. In addition, a difference in the abundance of Bacteroidota was observed between the two antibiotic-treated groups, with a higher abundance in UBRORED than CEF-treated colostrum (*p* < 0.05). Similarly, the lowest abundance of Patescibacteria was observed in both the antibiotic-treated groups when compared with the NOAB group (CEF *p* < 0.0001; UBRORED *p* < 0.001) and a significantly lower abundance was observed in the CEF-treated samples when compared with the UBRORED group (*p* < 0.01). These results demonstrate that colostrum from cows receiving no antibiotics during the dry period (NOAB) display a distinct microbial composition compared with both antibiotic-treated groups, characterized by lower Firmicutes and higher Bacteroidota. In addition, CEF and UBRORED colostrum also differ from each other across multiple taxa, including higher Bacteroidota in the UBRORED group, indicating that choice of DCT strategy can influence the microbial community of colostrum.

The choice of DCT also influenced the microbial composition at genus level. A total of 227 genera were identified in bovine colostrum and unknown sequences accounted for 24%. The eight most predominant genera, representing 40% of the total relative abundance, were identified as *Acinetobacter*, *Corynebacterium*, *Facklamia*, *Jeotgalicoccus*, *Lactococcus*, *Leuconostoc*, *Psychrobacter*, and *Staphylococcus* (Figure 3).

### 2.2. Differential Abundance Analysis

To examine genus level changes associated with antibiotic treatment, differential abundance analysis was performed with MaAslin2. Of the 227 genera detected in bovine colostrum, 224 were differentially abundant when the effect of antibiotic treatment was considered with NOAB as the reference group. A total of 111 and 113 features (genera) were significant in the CEF and UBRORED group, respectively. A positive coefficient indicates a higher abundance of a genus in the specified antibiotic-treated group, whereas a negative coefficient reflects a lower abundance relative to the reference (NOAB group). To determine which features may be causing differences between DCT treatment groups, a heat map of the top 50 differentially abundant genera was created (see Figure 4). Of the top 50 features, *Rhodococcus*, *Paracoccus,* and *Sphingomonas* had increased abundance in both of the antibiotic-treated groups when compared with the NOAB group.

Differential analysis of the ten most dominant genera further identified significant genus level shifts associated with antibiotic treatment (Table 1). *Corynebacterium* was most abundant in the UBRORED group and was significant when compared with the NOAB group (β = 1.327, q = 0.0251). Similarly, *Facklamia* and *Jeotgalicococcus* were elevated in UBRORED relative to NOAB (β = 1.36, q = 0.0184 and β = 1.580, q = 0.0241, respectively). *Staphylococcus* showed the lowest abundance in NOAB, differing significantly from both the CEF (B = 3.002, q value = 0.003) and UBRORED (β = 2.546, q = 0.0115) groups. *Leuconostoc* was significantly enriched in the CEF group when compared with NOAB (β = 3.766, q = 0.0099). No significant differences were detected for *Acinetobacter*, *Lactococcus,* or *Psychrobacter* across the DCT treatment groups (q > 0.05). Differential analysis between the antibiotic-treated groups revealed no significant associations among the ten dominant genera when comparing CEF and UBRORED directly.

In addition, a number of mastitis-associated genera were significantly different depending on antibiotic treatment received. In the CEF group, the abundance of *Pseudomonas* (β = 2.593, q = 0.0017), *Staphylococcus* (β = 3.002, q = 0.0033), and *Enterococcus* (β = 2.167, q = 0.0309) was higher when compared with NOAB (Figure 5). Similarly, in the UBRORED samples, an increased abundance of *Corynebacterium* (β = 1.327, q = 0.0251) and *Staphylococcus* (β = 2.546, q = 0.0115) was observed (Figure 5). Furthermore, direct comparison of the antibiotic-treated groups revealed significantly lower abundance of *Pseudomonas* in the UBRORED group compared with CEF (β = −3.8312, q = 2.9667).

### 2.3. The Influence of Dry Cow Therapy on the Microbial Diversity of Bovine Colostrum

#### 2.3.1. Alpha Diversity

The alpha diversity indices Chao1 and Shannon were used to determine intra-sample variation in all three DCT treatment groups (see Figure 6). The Chao1 index assesses the richness of bovine colostrum samples, whereas the Shannon index estimates both richness and evenness. According to both measures, the highest α-diversity was observed in the NOAB group (NOAB vs. CEF: Chao1 *p* < 0.01, Shannon *p* < 0.01; for NOAB vs. UBRO: Shannon *p* < 0.05). Furthermore, comparison of the diversity indices indicates that the microbiota within the CEF group has lower species richness (CEF vs. UBRO: Chao1 *p* < 0.05) and less diversity (Shannon *p* < 0.01) in comparison with colostrum from the UBRORED group.

#### 2.3.2. Beta Diversity

Beta diversity was used to measure dissimilarity in the microbial diversity of colostrum between the three DCT treatment groups. To determine if significant differences in β-diversity existed, a PCoA plot based on the Bray–Curtis distance matrix was constructed and PERMANOVA was performed. As shown in Figure 7, β-diversity analysis revealed the distinct clustering of the colostrum microbiota in the NOAB group. PERMANOVA showed the β-diversity of the NOAB group was significant against the CEF (*p* = 0.001) and UBRORED (*p*= 0.001) groups. In addition, discrete clustering of the colostrum microbiota between the two antibiotic-treated groups occurred and significance was confirmed (CEF vs. UBRORED *p* = 0.001).

### 2.4. The Influence of Individual Farm Variation on Bovine Colostrum

Alpha diversity indices Chao1 and Shannon revealed significant variation in microbial richness and diversity at the farm level (Figure 8). The highest α-diversity was observed in farm C. Chao1 estimated richness was significantly higher in farm C when compared with farm D (*p* < 0.0001) and farm P (*p* < 0.01). The lowest α-diversity was observed in farm D. The Chao1 index for farm D was significantly lower to farm L (*p* < 0.001), farm P (*p* < 0.01), and farm T (*p* < 0.05). Furthermore, using the Shannon index, significant variations in microbial richness and uniformity were observed in all five farms and significance was confirmed with the Wilcoxon test. With regards to β-diversity, PCoA based on the Bray–Curtis distance matrix revealed colostrum microbiota differs significantly according to farm. As depicted in Figure 9, discrete microbial grouping occurred between all five farms and significance was obtained with PERMANOVA (Farm C vs. all farms *p* < 0.01; Farm D vs. Farm L *p* < 0.001; Farm D vs. Farm P *p* < 0.01; Farm D vs. Farm T *p* < 0.05).

### 2.5. Parity and the Microbial Composition of Bovine Colostrum

When colostrum samples were grouped according to their parity status (0, 1, 2, 3, 4+), α-diversity metrics revealed no significant differences in richness and evenness (*p* > 0.05) (Figure 10). Similarly, β-diversity analysis showed no distinct grouping or significance in the microbial diversity between colostrum samples when grouped according to parity (*p* > 0.05) (Figure 11).

## 3. Discussion

Extensive data are available on the microbial profile of bovine milk [60]; however, less is known on the influence of antibiotic treatment on bovine colostrum, particularly from healthy cows. Current EU legislation prohibits prophylactic antibiotic therapy in dairy cows [47], which has raised concerns amongst farmers for animal health during lactation [61]. In the present study, we sought to not only identify the microbial composition of bovine colostrum in Irish dairy cows but also assess the effect of antibiotic administration. The samples used in this study were collected prior to the implementation of the ban on Blanket DCT (February 2020). Thus, this study examined the influence of three methods of DCT on the microbiota of bovine colostrum. Finally, we also accounted for the influence of additional factors, including parity and inter-farm variation.

Our study confirms the presence of a highly diverse and rich microbial population in bovine colostrum that is subject to variation with DCT. In this study, the abundance of several phyla was established, with Firmicutes (48%), Actinobacteria (22%), Proteobacteria (21%), and Bacteroidota (7%) representing the predominate phyla, regardless of the chosen DCT method. Thus far, there has been broad agreement at the phylum level, and our findings align with previous studies [16,33,44,45,62]. The distribution of the predominant phyla in both milk and colostrum has been shown to vary with Proteobacteria, Firmicutes, Bacteroidetes, and Actinobacteria, accounting for 23–84%, 16–68%, 3–25%, and <0.1–8%, respectively [25,63,64,65]. Although variations in the relative frequencies of the dominant phyla have been established, the driving cause for such fluctuations is not fully understood. In our study, the phylum Firmicutes was dominant across all treatment groups. However, the relative abundance of Firmicutes was significantly lower in the NOAB group as opposed to both antibiotic-treated groups. Similarly, Bacteroidota was influenced by antibiotic administration, with the highest relative abundance in the NOAB group. Furthermore, the choice of antibiotic proved significant, with increased relative abundance in the UBRORED group. Contrarily, a previous study noted no change in Bacteroidota abundance at the phylum level when comparing DCT with a teat sealant alone or a teat sealant with antibiotic treatment [66]. The lower abundance of Firmicutes in the NOAB group is unexpected and the cause remains unclear. However, this may be a result of a more balanced community profile, allowing competing Bacteroidota to occupy a greater proportion of the colostrum microbiota. Additionally, farm level factors may contribute to the observed pattern, as Firmicutes are known to vary with housing, hygiene, and farm management conditions. Previous studies suggest that the relative proportions of phyla may be an indicator of the health status of the cow. For instance, an increased relative abundance of Proteobacteria was established in healthy cows, whereas in cows with subclinical mastitis, Firmicutes was predominant [65]. Moreover, Kaczorowski et al. suggest that the ratio of phyla abundance is key, with Firmicutes and Proteobacteria occurring in approximately equal amounts in a healthy cow [67], whereas the occurrence of subclinical mastitis creates a shift, dependent on the causative pathogen, allowing one phylum to dominate. Indeed, other studies noted a similar trend; Proteobacteria was dominant in mastitis caused by *E. coli* (98%), whereas Firmicutes had higher abundance in *Streptococcus*-associated mastitis (69.6%) [68]. However, in healthy cows, the accuracy of the equal ratio remains unclear. Indeed, the same study noted abundances of 57.7% and 26% for Firmicutes and Proteobacteria in healthy dairy cows, respectively. These proportions are similar to those observed in our study, where the cohort of experimental cows were infection-free. Evidently, our findings substantiate existing data on the predominant phyla in bovine colostrum but also highlight fluctuations in their proportions as a result of antibiotic administration during drying off. The potential influence of such changes in relative abundance at the phylum level on ruminant health remains unclear.

In previous studies, much greater variation in relative abundances at lower taxonomic levels has been demonstrated, including genus. Of the wide variety of bacterial genera identified in bovine colostrum and milk, some include members with known probiotic, spoilage, and infectious properties. Commonly identified genera include *Acinetobacter*, *Corynebacterium*, *Staphylococcus*, *Streptococcus*, *Pseudomonas*, and *Escherichia shigella* in both bovine colostrum [16,25,33,44,64,69] and milk [65,67,70,71]. In our study, the predominant genera were *Psychrobacter* (10.75%), *Corynebacterium* (6.21%), *Facklamia* (6.06%), *Jeotgaliococcus* (4.21%), *Staphylococcus* (4.08%), *Lactococcus* (3.77%), *Acinetobacter* (2.96%), and *Leuconostoc* (2.34%). Unknown genera accounted for 23.86% and all other genera had an abundance of less than 2%.

We observed significant variations in the abundance of different genera as a result of DCT treatment. Our study demonstrated increased abundance of the *Facklamia* genus as a result of UBRORED administration when compared to NOAB-treated samples. Members of the genus have previously been associated with the bovine female genitourinary tract [72] and identified as a predominant member of the teat apex, cistern, and canal of healthy dairy cows [73,74]. Recently, *Facklamia* was identified as a predominant genus in raw bulk tank milk and an increased abundance was associated with high feed efficiency [75]. Additionally, two novel strains were identified in raw bulk tank milk, suggesting *Facklamia* may be a putative member of the milk microbiota [76]. Elsewhere, a study noted the highest abundance of *Facklamia* was associated with subclinical mastitis, followed by healthy cows, and lastly cows with clinical mastitis [77]. Overall, the genus is regarded as a commensal bacteria in the bovine milk microbiota [78]. However, its isolation from infected sources demands further exploration of its potential pathogenesis. Similarly, an increased abundance of *Jeotgaliococcus* was observed in the UBRORED group when compared with colostrum following natural DCT (NOAB). This genus was first isolated from a traditional Korean fermented seafood, jeotgal [79], and has since been identified in a wide range of sources including a poultry house, marine animals and environments, urban air samples, and goats milk [80,81,82,83,84,85]. *Jeotgaliococcus* was also identified in the bovine teat canal and on the bovine foot skin [73,86]. Recently, the genus has been identified in both healthy and subclinical mastitis milk [87]. Little is known of its functional role in milk, specifically regarding the potential pathogenesis and resistome profile of the genus, although resistance to heavy metals has been demonstrated [88]. Our results are consistent with *Facklamia* and *Jeotgaliococcus* being commensal bacteria of bovine colostrum, with increased abundance as a result of DCT with β-lactam antibiotics of the penicillin class.

In agreement with our findings, *Staphylococcus* has previously been identified as a dominant genus in milk samples from healthy cows, as well as subclinical and clinical mastitis milk samples [52,67,89]. Staphylococci are a primary cause of mastitis in dairy cows worldwide [77], with mastitis-associated staphylococci divided into two groups: *S. aureus*, a major mastitis causing pathogen, and coagulase negative (CNS) staphylococci, considered opportunistic mastitis pathogens, for example, *S. chromogenes*, *S. epidermis*, *S. haemolyticus*, *S. simulans,* and *S. xylosus* [90]. Both groups of *Staphylococcus* have been identified in milk from subclinical mastitis, clinical mastitis, and healthy cows [77,89,91,92]. Furthermore, staphylococci have been isolated from extra-mammary sites including the perineum skin, udder skin, teat apices, and teat canal of lactating cows, in addition to the farm environment [91,93,94]. In this study, the abundance of *Staphylococcus* was increased as a result of antibiotic treatment, regardless of the class of antibiotic administered. In contrast, a previous study noted the abundance of *Staphylococcus* was reduced following antibiotic treatment in cows with subclinical and clinical mastitis [77]. The reason for increased abundance of *Staphylococcus* with antibiotic treatment in our study is unclear; however, it may be a result of disruption to the natural community of microbes in colostrum, facilitating increased *Staphylococcus* dominance. Despite the acknowledged existence of staphylococci as a commonly occurring microbe in colostrum and milk, its increased prevalence following antibiotic administration during DCT is a concern and warrants further research, due to the pathogenic nature of various members of the genus. The *Staphylococcus* genus is known to have both intrinsic and acquired virulence factors that contribute to their efficiency as a pathogenic organism and challenge the dairy industry at controlling their spread [90,95]. In addition, further contributing to this challenge and increasing the risk to animal welfare is the rising resistance to β-lactam antibiotics observed in both *S. aureus* and CNS *Staphylococcus* strains [90,96].

The highest prevalence of the genus *Corynebacterium* was observed in the UBRORED group, with a significantly lower abundance in the NOAB group, although the same was not observed in the CEF-treated colostrum. *Corynebacterium* is frequently isolated from milk, with members of the genus having both pathogenic and non-pathogenic roles. The prevalence in bovine milk is believed to be due to contamination on farms, facilitating the uptake of opportunistic pathogenic members and non-pathogenic colonizers. Previously, *Corynebacterium* was identified as a dominant genus in milk from Holstein dairy cows, with an increased abundance in winter milk samples when compared with summer sampling [75]. This may explain its high abundance across all samples in our study, as sample collection occurred following winter housing months. The genus *Corynebacterium* represents species frequently associated with mastitis, most notably *C. bovis*. Indeed, *C. bovis* was identified as the dominant species in subclinical mastitis milk and was associated with an increase in SCC [97]. Similarly, *C. bovis* was the predominant species in both healthy and infected samples and associated with an increased SCC [98]. Although *C. amycolatum* was also identified, it is believed to be less clinically relevant [98]. Likewise, other members of the genus were infrequently isolated in subclinical mastitis samples, and were not associated with increased SCC [97]. Indeed, in addition to the presence of the genus in clinically infected milk, it has also been established in healthy milk [67,77,98]. These findings indicate that although *C. bovis* is likely the predominant species from the genus in bovine milk, other members of the genus are present and their pathogenicity is unknown. Interestingly, a previous study noted increased levels of *Corynebacterium* following antibiotic treatment in both clinical and subclinical groups compared with pre-antibiotic administration [77]. Additionally, a lower abundance of *Corynebacterium* in clinical mastitis samples both before and after treatment was observed when compared with samples from the healthy cohort. *Leuconostoc* are facultative anaerobic lactic acid bacteria with a well-established role in food fermentation and technology [21,99]. They are commonly found in plants, dairy products, meats, and fermented foods [26], and are frequently identified in bovine milk [45,100]. Several species, including *L. mesenteroides* and *Leuconostoc lactis*, have been identified in bovine milk [101]. Despite their technological value, members of the genus have previously exhibited high levels of resistance to vancomycin [102]. Furthermore, multiple *Leuconostoc* spp. have demonstrated multi-drug resistance, including resistance to tetracycline, chloramphenicol, trimethoprim, and vancomycin [103]. The increased abundance of *Leuconostoc* in the CEF-treated colostrum is unexpected; however, it may reflect suppression of more susceptible taxa by cefquinome, enabling *Leuconostoc* to persist or proliferate. Notably, this pattern was not observed in the UBRORED group, suggesting a treatment-specific effect. The potential antimicrobial resistance traits of *Leuconostoc* warrant further investigation. Future work should evaluate any implications for udder or calf health and characterize the antimicrobial-resistant capacities associated with *Leuconostoc*. We observed a high degree of commonality in the abundance of *Acinetobacter,* regardless of DCT approach. The high abundance of *Acinetobacter* in colostrum is not surprising, as it is commonly found in nature and frequently identified in soil and water-based ecological niches [104]. The genus has previously been identified in milk from healthy [67], subclinical [67,77], and clinical mastitis [56] cows. Of the genus, *Acinetobacter baumannii* has garnered the most attention due to its role in nosocomial infection, intrinsic antibiotic resistance, and ability to adapt to unfavourable environmental conditions [105]. Its presence in milk is linked with environmental contamination via the farm. Indeed, *A. baumannii* isolates were present in 21.2% of 280 dairy cows and identified on the bovine nasal area, the farm floor, and rectal swabs [106]. However, the genus consists of over 80 species [107]. Indeed, in bulk tank milk, 176 *Acinetobacter* species were isolated, with *A. baumannii* accounting for only 32% [108]. Separately, 27 strains of *Acinetobacter* were isolated from milk [56]. Evidently, colostrum is host to a highly diverse population of *Acinetobacter*. Although members of the genus have previously shown resistance to antibiotics of both the cephalosporin and penicillin class, their susceptibility has also been demonstrated [106,108]. The high abundance in the NOAB group and persistent abundance in the antibiotic-treated group is likely due to the role of *Acinetobacter* as an innocuous component of the bovine milk microbiota. In contrast to the findings presented here, a previous study noted highest levels of *A. baumannii* isolation from milk when a cephalosporin had been used on the farm in the preceding six months [106]. Despite a growing understanding of the pathogenic members of the genus, little remains known of the non-pathogenic species. A recent review suggests the dualistic nature of *Acinetobacter*, acknowledging the pathogenicity of some whilst highlighting the potential role of the genus in biotechnological processes [104]. Evidently, further research is required of *Acinetobacter* species identified in milk to fully ascertain their role as part of the wider bovine colostrum microbiota network. Other predominant genera identified in our study include *Psychrobacter* and *Lactococcus*; no significant variations in their relative abundance occurred as a result of DCT.

Importantly, in our study, genera commonly associated with mastitis, including *Staphylococcus* and *Corynebacterium*, were higher in colostrum samples following blanket DCT than natural DCT. Additionally, a positive association was observed in *Pseudomonas* and *Enterococcus* abundance in the CEF group following differential analysis, with NOAB as the reference. Both *Pseudomonas* and *Enterococcus* have previously been identified as genera representing strains with pathogenic capabilities, namely *P. aeruginosa* and *E. faecalis*, respectively [109,110,111]. Evidently, the treatment of cows with antibiotics during DCT does impact the microbial composition of colostrum, with an increase in mastitis-associated genera observed. This may be due to the presence of other non-pathogenic species belonging to the same genus but could also indicate a disruption to the microbial population, resulting in increased abundances of genera with known virulence factors. Future studies should investigate the observed disruption by employing techniques such as shot-gun sequencing, which will allow for greater understanding of strain level disruption and the implication for long-term animal health and productivity. In particular, the potential presence of antibiotic-resistant bacteria and genes should be investigated. Critically, natural DCT did not result in increased pathogenic microbes, suggesting its potential as a farm management strategy.

Studies to date indicate a higher α-diversity for healthy milk samples and lower levels demonstrated in mastitis milk [44,70,112,113,114]. This is likely the result of dominance by the causative pathogenic strain during infection, although the opposite has also been reported [78,89]. In the majority of studies to date, α-diversity following antibiotic administration was assessed in mastitic cows. For example, no significant variation was observed in the α-diversity of bovine milk samples from subclinical and clinical mastitis groups, in receipt of antibiotics, when compared with untreated healthy milk samples [77]. Furthermore, when comparing a mastitis group in receipt of antibiotics with a mastitis positive group not receiving antibiotic therapy, no difference in α-diversity was observed [114,115]. However, the effect of antibiotics alone cannot be determined due to the presence and potential influence of infection status, whereas in this study milk was collected from healthy cows who received prophylactic antibiotic administration. However, two previous studies examined the Chao1 and Shannon indices of colostrum samples from a cohort of healthy cows following antibiotic treatment or teat sealant alone and found no significant difference [63,71]. These findings are in contrast to those presented here. We identified a significant increase in microbial richness and evenness in colostrum samples from cows that received no antibiotic treatment during DCT, whilst the lowest was seen in the CEF-treated group. With regards to β-diversity, a previous study noted no significant difference between milk samples of infected cows in receipt of a broad spectrum antibiotic and the healthy group [77]. Furthermore, in a study similar to our own, β-diversity was not significant between those in receipt of antibiotics and those with teat sealant alone in a healthy cohort of cows [71]. Contrarily, for β-diversity analysis, we demonstrate a clear separation of the microbial diversity of the ‘NOAB’-treated colostrum samples from those following either UBRORED or CEF administration during the dry period. Additionally, a longitudinal study carried out by our group on a subset of the same samples analysed in this study demonstrated that distinct microbial clustering of the NOAB group persisted in samples even at 2 months post-DCT [55]. This emphasises the long-lasting effects antibiotic administration can have on the microbial profile of bovine colostrum and mature milk. Furthermore, the same study noted an increased abundance of antibiotic-resistant genes in bacterial strains following CEF and UBRORED DCT. These findings lend to the growing concern of the rapid emergence and proliferation of antimicrobial resistance due to antibiotic therapy during the drying off period. Our findings indicate that colostrum samples from cows that are treated naturally during DCT have a compositionally distinct microbiota to those following antibiotic therapy. Despite the high degree of commonality observed at phylum and genus levels, antibiotic administration does shift the microbial composition of colostrum. The separate grouping between treatment groups suggests potential dysbiosis in the microbial profile of bovine colostrum as a result of antibiotic administration. Dysbiosis has been suggested as a facilitating factor in the development of intramammary infections; however, whether mammary dysbiosis is the result of infection or the causative link remains unclear [20,22,23]. In addition, we also identified significant dissimilarity in microbial diversity between the two antibiotic-treated groups, Cephagaurd and Ubro Red, suggesting that the choice of antibiotic can further impact the colostrum microbiota profile. In this instance, both are broad spectrum β-lactam antibiotics. Cephagaurd is known to target mastitis-causing agents, including *S. aureus*, CNS *Staphylococcus,* and *Streptococcus* spp., including *S. uberis*, *S. dysgalactiae,* and *S. agalactiae.* Similarly, UBRORED also targets *Staphylococcus* and *Streptococcus* spp., in addition to *Corynebacterium*, *E. coli*, *Klebsiella*, and *Pseudomonas*. Bacteria exist within an interconnected and diverse microbial network. Fluctuations in the presence of one species can modify the functionality of the community, influencing the host and surrounding environment [94]. Further analysis of microbial disparity at the strain level will allow for a greater understanding of the influence of antibiotic therapy on bovine colostrum.

It is important to recognize that antimicrobials are a valuable tool in the treatment of disease amongst animals and humans. However, indiscriminate overuse has driven the need for a reduction in antimicrobial administration. The resulting EU ban on blanket DCT has led to the routine implementation of selective DCT on dairy farms. Previous studies have demonstrated the efficacy of selective DCT in treating and limiting the spread of infection, reducing the use of antimicrobials, without compromising animal health [116,117]. Similarly, teat sealant alone (natural DCT) has proven safe in low infection risk animals [66,118], further supporting the cessation of prophylactic antibiotic administration. Despite such findings, scepticism persists, with concerns for long-term animal health and productivity, in addition to potential antibiotic requirements during lactation [117]. Alternative therapies have been extensively reviewed previously [111,119]. Of these, *L. lactis* DPC3147 proved equally effective to conventional antibiotic therapy in treating subclinical and clinical mastitis [120] through the stimulation of the host intramammary immune response [121,122]. A variety of probiotic-based mitigation strategies are now being explored. For example, a *Lactobacillus*-based teat dip proved effective in reducing mastitis-associated bacteria, suggesting an alternative to current chemical-based teat disinfectants [123]. In addition, bacteriocins are proposed as promising antimicrobials to be included in teat seal and udder preparation formulations for mastitis prevention. The potential of a germicidal sanitizer containing Nisin (AMBICINN^®^) was previously demonstrated against mastitis agents and was comparable to an iodine teat dip [124]. Indeed, WipeOut dairy wipes (ImmuCell Corporation) are now commercially available. Impressively, the inclusion of lacticin 3147 in teat seal formulations proved effective for mastitis prevention [125,126,127,128]. Furthermore, an intramammary infusion of Nisin Z had therapeutic effects in cows with subclinical mastitis, with a high recovery rate in *S. agalactiae* cases, although lower recovery was observed in *Staphylococcus*-based infections [129]. Other novel therapies currently under investigation include plant-derived compounds, postbiotics, and bacteriophage therapy [130,131,132]. Evidently, the development of alternative treatment and mitigation options has received considerable interest, although further establishment of their efficacy is still required. The development and implementation of novel therapies, in addition to a reduction in on-site antibiotic use, will ensure the continued efficacy of antibiotic therapy and poses a reduced threat at farm and environmental levels.

Parity exerted no impact on the microbial composition or diversity of bovine colostrum in our study and others [16]. An increase in microbial richness (Chao1 index) of colostrum from primiparous cows has been reported elsewhere [44]. However, when analysed for richness and evenness (Shannon index), no significance was observed. Separately, no significant difference in α-diversity of primiparous and multiparous cows was observed; however, β-diversity analysis revealed distinct clustering of bovine milk from multiparous cows in early lactation [45]. Additionally, unlike our findings, both studies noted changes in relative abundances at the genus level. Despite conflicting reports regarding parity and the microbial composition, greater consensus exists in relation to parity and its influence on colostrum quality, determined by colostral Ig levels. Indeed, colostrum quality is known to increase with higher levels of parity [13,37,39,42]. In fact, a number of studies have noted that Ig content is highest in parity levels >3 [6,16,40,43]. This is most likely a result of the more mature and experienced immune system in comparison to the younger naïve immune system. Furthermore, nutritional components may also be influenced by parity, with higher fat and lactose levels in lower parity cows [13]. In contrast to microbial studies, findings to date are in agreement that parity does significantly affect the nutritional composition of colostrum; thus, it is likely that parity has greater influence on colostrum’s nutritional and bioactive components rather than the microbial population.

At a farm level, significant variation in Shannon index and β-diversity was observed, with significance demonstrated between all farms. Antibiotic therapy is likely the primary cause for the observed differences; however, significance was still observed in α-diversity (Shannon index) between farm D and P, despite the same antibiotic (Cephagaurd) being used. Similarly, for β-diversity, despite the same choice of antibiotic, differences were observed between farm D and P (Cephagaurd) and farms L and T (Ubro Red). Therefore, our findings suggest that additional factors influence the microbial profile of bovine colostrum. As we can account for breed, parity, and season, we know these factors are not the cause in our study (see Appendix A). However, we suggest that farm management practices, the farm environment, and choice of feed may explain the difference found between the dairy farms evaluated. Indeed, a previous study of colostrum from farms in China noted significant variations in the microbial structure of colostrum between two farms at the genus level, despite a high degree of commonality in the predominant phyla [33]. Similarly, the group suggests that feed and farm management practices were responsible for the observed differences. Overall, we note variation in microbial diversity between farms, in addition to the influence of antibiotic therapy. However, further investigation of genetic, environmental, and management factors need to be performed to determine the degree of influence on the bovine colostrum microbial population.

A key limitation in this study is the use of 16S rRNA sequencing at a single time point. Future studies should incorporate longitudinal sampling and higher resolution sequencing approaches. Shotgun metagenomics or whole-genome sequencing (WGS) would provide more detailed insights into the influence of DCT, strengthen the data on on-farm antibiotic usage, provide greater insight into the potential resistome of bovine colostrum, and enable a greater understanding of species level variability and functionality. Identifying the source of ARGs on dairy farms is critical to mitigate the global spread of antibiotic resistance. Further investigations are needed to determine the origin and mechanism of transmission, including horizontal and vertical gene transfer. To this end, the inclusion of both environmental (bedding, water, soil) and faecal samples would provide a more comprehensive insight into resistome dynamics on farms. In addition, quantitative techniques such as flow-cytometry-based methods, including Flow-FISH, could complement sequencing data by providing direct measurements of absolute bacterial loads in colostrum. Furthermore, the inclusion of an additional treatment group of cows with a positive infection status would indicate how infection modulates microbial responses to drying off. Future studies should monitor SCC and IgG levels to confirm udder health and evaluate any secondary effect of DCT on colostrum quality. IgG testing can be performed on-farm using a Brix refractometer; however, for improved accuracy, laboratory-based methods such as the radial immunodiffusion assay or ELISA can be performed [133,134]. For SCC testing, several on-farm techniques are available, including the California Mastitis Test, the Porta SCC test, and the Delaval Cell Counter [135]. In addition, an automated fluorescence-based cell counter, the Fossomatic, is widely used and provides high-throughput, accurate results. Finally, across microbiota studies to date, a number of different sampling, DNA extraction, and sequencing methods have been employed, potentially introducing variations in the data gathered following milk microbiota analysis.

This study provides new insight into how different DCT strategies influence the microbial composition of bovine colostrum under commercial dairy farm conditions. Our findings demonstrate that natural drying off is associated with a distinct colostrum microbiota, without an increased abundance of mastitis-associated genera. These results have practical implications for farm management practices, supporting the current move towards reduced prophylactic antibiotic administration and highlighting the need for further development of targeted non-antibiotic alternatives to maintain udder health.

## 4. Materials and Methods

### 4.1. Study Population

Colostrum samples were collected from five dairy farms in the same locality of North Cork in February 2020, with an approximate herd size of 75 cows per farm. Experimental cows were selected at random and any cows presenting signs of mastitis or other infection during the dry or sampling period were excluded. A total of 90 Holstein Friesian cows were selected for this study, providing a total of 90 colostrum samples. The farms included in this study were identified as Farm C, Farm D, Farm L, Farm P, and Farm T. Farms enrolled in this study on a voluntary basis and no changes in the existing farm management practices were made due to participation in this study. All cows were housed indoors during the winter until calving (November-February). On all farms, cows were treated with teat sealant (Boviseal, Buyrite Solutions, Co., Wexford, Ireland). In addition, four of the five farms administered intramammary antibiotics to multiparous cows at the start of the drying off period (blanket DCT). Two of the farms (L and T) used ‘Ubro Red Dry Cow Intramammary Suspension’ and two of the farms (D and P) used ‘Cephaguard DC 150 mg intramammary ointment’ for drying off. Ubro Red contains the active ingredients Framycetin Sulfate (100 mg), Penethamate Hydriodide (100 mg), and Procaine Penicillin (300 mg). For Cephaguard, the active ingredient is Cefquinome (150 mg), a fourth-generation cephalosporin. In order to evaluate the influence of antibiotic therapy, colostrum samples from these farms were designated as UBRORED group and CEF group, respectively. One farm (C) did not use any antibiotic therapy (natural drying off) and was grouped as the no antibiotic (NOAB) group. The NOAB group also included samples from primiparous cows of farms D, L, P, and T. For parity analysis, colostrum samples were grouped according to the number of previous gestations: 0 (first gestation), 1, 2, 3, 4+. See Appendix A for metadata on the experimental cows and farms used in this study.

### 4.2. Sample Collection

All farms and personnel who participated in sample collection were quality-assurance-certified members of the Irish Food Board (Bord Bia). Farmers completed a questionnaire for each sample, providing information on the identification number, age, infection status, antibiotic history, parity of the cow, and the date of calving. All farmers were provided with and followed instructions for sterile sample collection, in accordance with the recommendations from the National Mastitis Council’s Laboratory Handbook on bovine mastitis [136]. In brief, colostrum samples were collected at the first milking within 1 h of calving. To ensure mammary gland stimulation, the first streams of colostrum from each mammary quarter were discarded (fore-stripping). Pre-dipping was then carried out by dipping the teats in iodine tincture. Teats were dried and scrubbed with wipes soaked in 70% alcohol. Finally, 15 mL of colostrum were collected in sterile falcon tubes (Starstedt, Nümbrecht, Germany). Each sample was labelled with the cow’s identification number and the date of calving. Samples were stored immediately at −20 °C in a chest freezer on farm for a maximum of one week. Finally, samples underwent cold storage and were transported to Teagasc, Moorepark and stored in a −80 °C freezer, awaiting analysis.

### 4.3. DNA Extraction

Prior to extraction, samples were removed from the −80 °C freezer and allowed to thaw at 4 °C overnight. Microbial DNA was extracted following a modified protocol for the DNeasy Powerfood Microbial kit (Qiagen, Manchester, UK), as previously described [137]. In brief, 12 mL of sample was transferred to a sterile falcon, homogenized by inverting multiple times and centrifuged at 4000× *g* for 30 min at 4 °C. The fat layer was removed using a sterile cotton swab (Thermo Fisher Scientific Inc., Waltham, MA, USA) and the supernatant containing protein was decanted. The remaining cell pellet was re-suspended in 1 mL of sterile phosphate-buffered saline (PBS) (Sigma Aldrich, St. Louis, MO, USA) and transferred to a sterile 1.5 mL tube and centrifuged at 13,000× *g* for 1 min at room temperature. The supernatant was discarded. This wash step was repeated until the supernatant was no longer cloudy. The pellet was re-suspended in 1 mL of PBS and treated with 90 µL of 50 mg/mL of lysozyme from chicken egg white (Sigma Aldrich) and 50 µL of 5 KU/mL Mutanolysin from *Streptomyces globisporus* ATCC 21553 (Sigma Aldrich). The samples were incubated at 55 °C for 15 min and bump vortexed at 3 min intervals. Samples were removed from heat and 28 µL of proteinase K (Sigma Aldrich) was added before further incubation at 55 °C for 15 min, followed by centrifugation at 14,000× *g* for 10 min at 4 °C. The supernatant was discarded. Finally, DNA extraction was completed using the DNeasy Microbial Powerfood Kit following the manufacturer’s instructions. A negative control using sterile molecular grade water (VWR International Limited, Co., Dublin, Ireland) was included. Following extraction, samples were stored at −20 °C, awaiting further analysis.

### 4.4. Preparation of DNA for Sequencing

PCR amplification of the V3-V4 hypervariable region of the 16S rRNA gene was carried out using the 16S metagenomic sequencing library protocol (Illumina, San Diego, CA, USA). For amplification, template DNA was amplified using specific V3-V4 region primers; forward primer 5′ TCGTCGGCAGCGTCAGATGTGTATAAGAGACAGCCTACGGGNGGCWGCAG; and reverse primer 5′ GTCTCGTGGGCTCGGAGATGTGTATAAGAGACAGGACTACHVGGGTATCTAATCC. The PCR reaction was composed of 2.5 µL of template DNA, 5 µL forward primer (5 µM), 5 µL reverse primer (5 µM), 12.5 µL 2× KAPA HiFi Hotstart ready mix (Anachem, Luton, UK), and molecular grade water (VWR). The applied PCR conditions were an initial denaturing step of 95 °C for 3 min, followed by 30 cycles of 95 °C for 30 s, 55 °C for 30 s, 72 °C for 30 s, and a final elongation step of 72 °C for 5 min. To confirm amplification, gel electrophoresis (1× TAE buffer, 1.5% agarose, 100 V for 30 min) was used for visualization of the PCR product. For purification of the amplicons, AMPure XP magnetic bead-based purification (Labplan, Co., Kildare, Ireland) was used, followed by a second PCR reaction. For indexing, dual indices and Illumina sequencing adapters (Illumina Nextera XT indexing primers, Illumina) were added to the DNA product (5 µL). Samples were quantified using the Qubit (Bio-Sciences, Dublin, Ireland) and the high specificity DNA quantification assay kit (Bio-Sciences). Following quantification, samples were pooled in equal molar concentrations. For quality analysis, the pooled samples were run on the Agilent Bioanalyser (Agilent Technologies, Santa Clara, CA, USA) and prepared for sequencing in accordance with Illumina guidelines. Samples were sequenced on the MiSeq sequencing platform in the Teagasc Sequencing facility, using a 2 × 300 cycle V3 kit, following standard Illumina sequencing protocols.

### 4.5. Bioinformatic and Statistical Analysis

The raw sequencing data were processed using the DADA2 pipeline (version 1.16) as previously described [138]. Specifications for optimal data refinement were applied, including maxN = 0, maxEE = c (2, 2), rm.phix = TRUE, truncQ = 2, trimLeft = c (17, 21), and truncLen = c (280, 210). The selected parameters ensured rigorous quality control and accurate sequence inference. Using the processed data, an Amplicon Sequence Variant (ASV) table was generated. Taxonomic rank was assigned using the SILVA v138.1 database [139]. Finally, to enable downstream analysis, a Phyloseq object was created through the combination of sample metadata and OTU table and taxonomic data [140]. Statistical analysis was performed using R (version 4.3.2). Statistical significance was accepted as *p* < 0.05. Relative abundances were calculated and visualized using the R software packages phyloseq, microbiomeutilities, microbiome, and ggplot2. Pairwise comparison was performed using the Wilcoxon signed-rank test with the command ‘geom_pwc’, and significant *p*-values were adjusted using the Benjamin Hochberg method. Alpha diversity (within sample variation) was determined using Chao1 and Shannon indices. Beta diversity (between sample variations) was measured based on the Bray–Curtis distance matrix and visualized using Principal Co-ordinate Analysis (PCoA). Significance was obtained using Permutational Analysis of Variance (PERMANOVA). Differential abundance analysis was performed at a genus level using the *MaAsLin2* package in R. The general linear model, log transformation, and CLR normalization were the applied parameters. Antibiotic-treated groups (UBRORED and CEF) were selected as fixed effects and NOAB was chosen as the reference group, unless stated otherwise.

## 5. Conclusions

Our data indicate no increased abundance in mastitis-associated genera with natural DCT. Mammary gland health is irrevocably linked with animal productivity, well-being, and end product quality. We suggest non-antibiotic alternatives could be a superior long-term solution due to increased bacterial dysbiosis and resistance associated with prophylactic antibiotic administration. Our study may have practical implications for farm management strategies during the dry period and lends support to the recent EU ban of blanket DCT. Rapidly evolving antimicrobial resistance is a global health concern and a reduction in the use of antimicrobials in dairy farming would have undeniable benefits both economically and from a One Health perspective; therefore, the future development of targeted novel therapeutic strategies is critical.

## Figures and Tables

**Figure 1 antibiotics-14-01217-f001:**
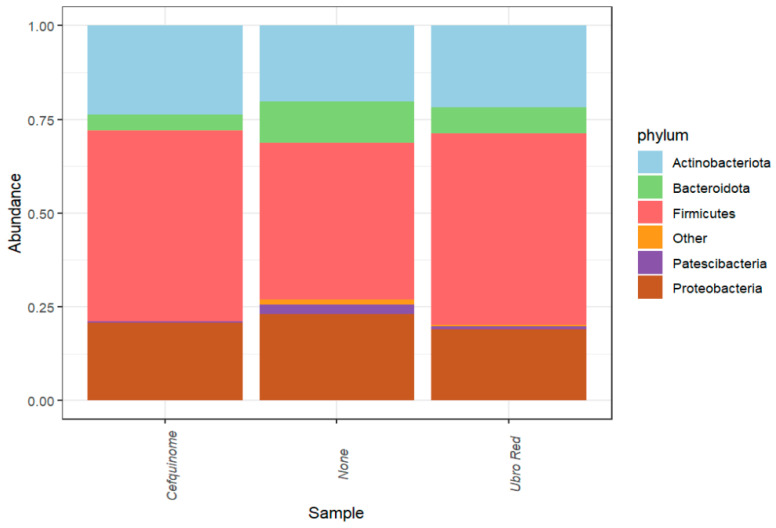
The 5 most prevalent phyla in bovine colostrum grouped according to antibiotic treatment.

**Figure 2 antibiotics-14-01217-f002:**
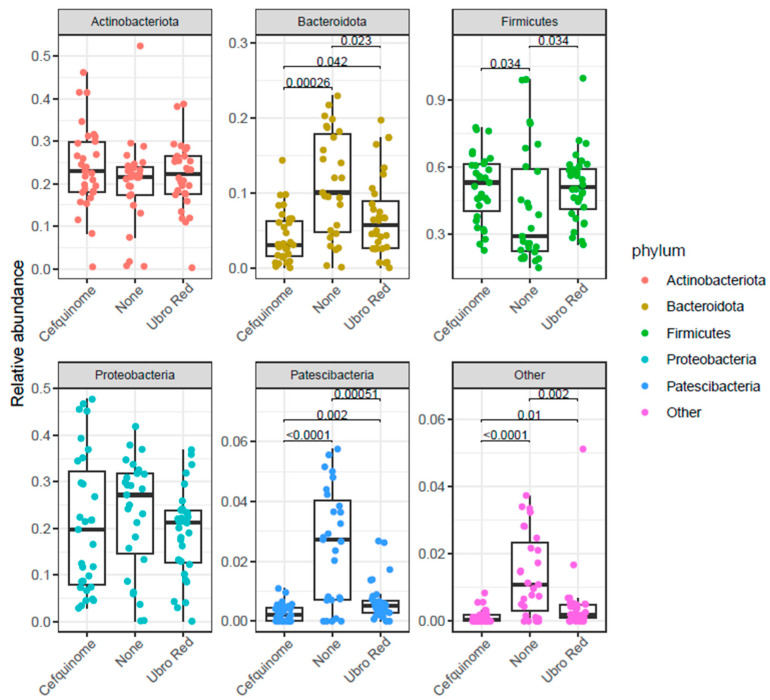
The distribution of abundance of phyla according to dry cow therapy. Significance was determined using ggpubr for pairwise comparison. The *p*.adj values < 0.05 were considered significant.

**Figure 3 antibiotics-14-01217-f003:**
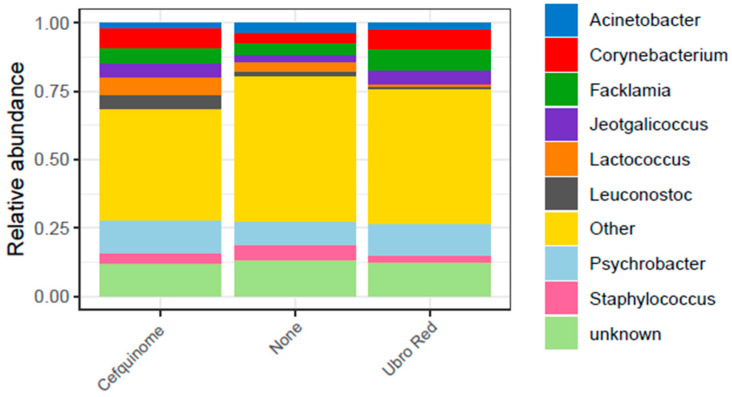
The relative abundance of the top 10 genera identified in bovine colostrum from all three DCT treatment groups.

**Figure 4 antibiotics-14-01217-f004:**
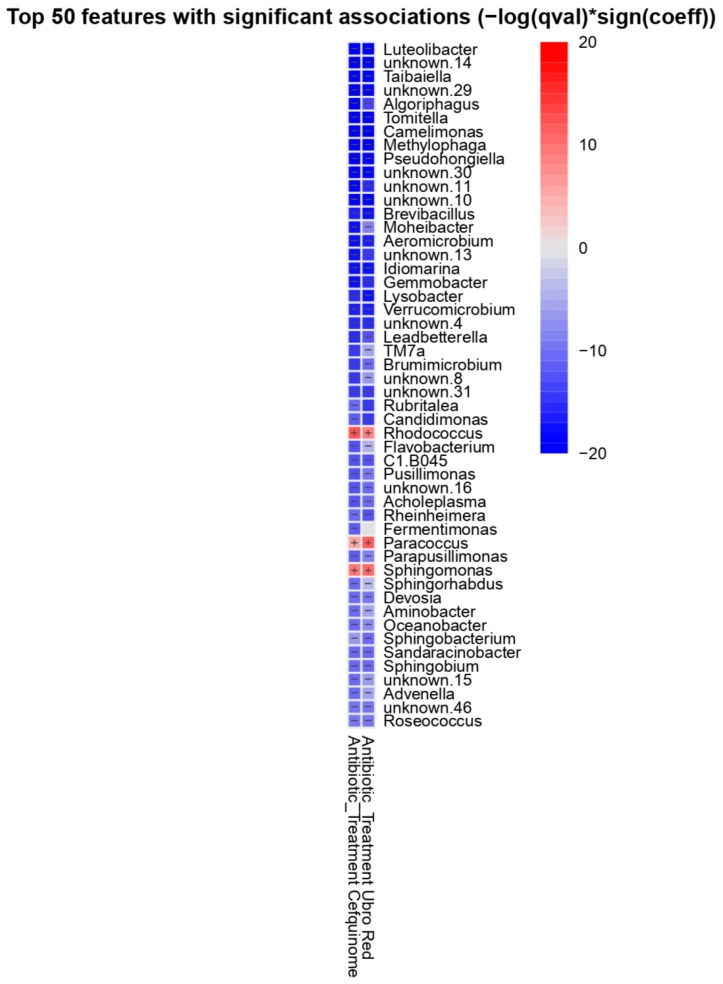
A heat map showing the top 50 taxa identified as differentially abundant in antibiotic-treated groups (CEF and UBRORED) relative to the NOAB reference group using MaAslin2. Blue tiles indicate a negative association (depletion relative to NOAB) and red tiles indicate a positive association (enrichment).

**Figure 5 antibiotics-14-01217-f005:**
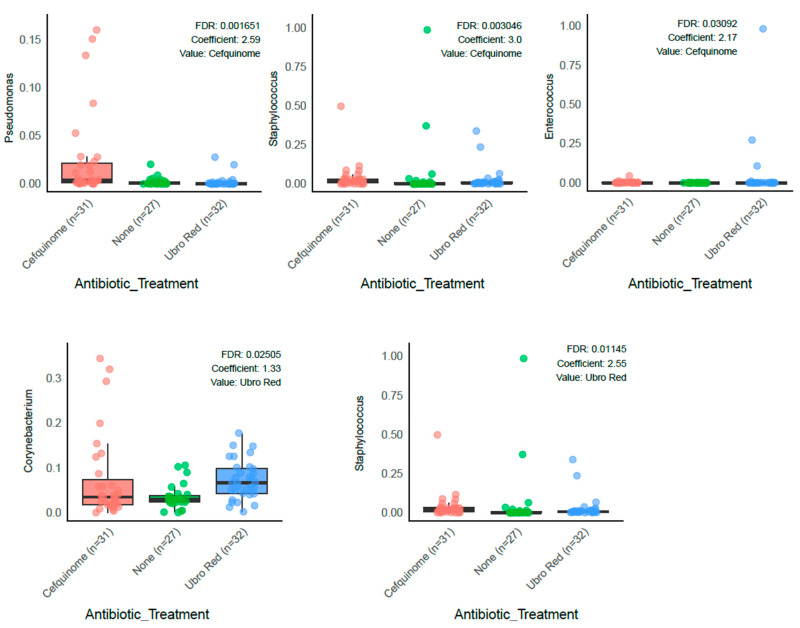
Differential abundance analysis of mastitis-associated pathogens between the antibiotic-treated groups and the NOAB group: *Pseudomonas*, *Staphylococcus,* and *Enterococcus* demonstrated higher abundance in the CEF group relative to the NOAB group. *Staphylococcus* and *Corynebacterium* were significantly enriched in the UBRORED group relative to the NOAB group.

**Figure 6 antibiotics-14-01217-f006:**
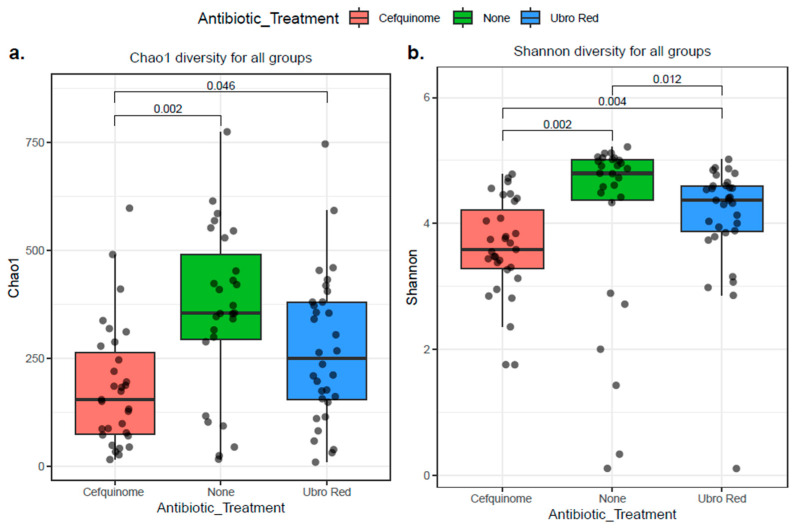
Alpha diversity indices were calculated for bovine colostrum samples grouped according to dry cow therapy treatment. (**a**) Chao1 index of species richness. (**b**) Shannon index of species richness and evenness. The *p*.adj values < 0.05 were considered significant.

**Figure 7 antibiotics-14-01217-f007:**
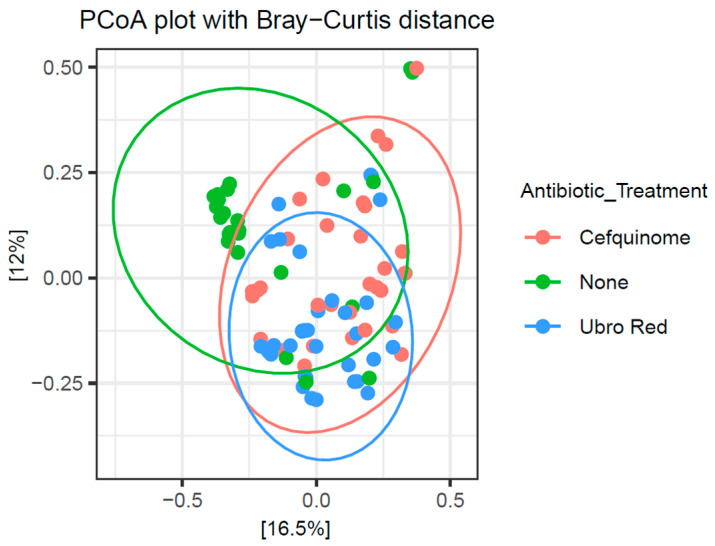
A PCoA plot based on the Bray–Curtis distance dissimilarity matrix of the inter-sample variation according to dry cow therapy treatment. The plot reveals the discrete clustering of the NOAB group when compared with CEF and UBRORED groups. A significant difference in the colostrum microbiota β-diversity of the NOAB group in comparison with both antibiotic-treated groups was determined with pairwise Adonis test (NOAB vs. CEF *p* = 0.001; NOAB vs. UBRORED *p* = 0.001). Separate grouping also occurred between the two antibiotic-treated groups and significance was confirmed (CEF vs. UBRORED *p* = 0.001).

**Figure 8 antibiotics-14-01217-f008:**
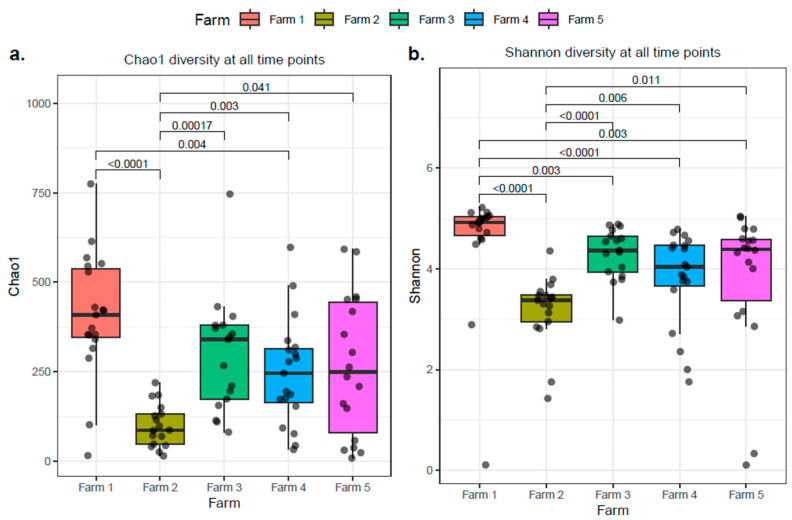
Colostrum samples grouped by farm showed significant differences in alpha diversity, as measured by (**a**) Chao1 and (**b**) Shannon indices. The *p*.adj values < 0.05 were considered significant.

**Figure 9 antibiotics-14-01217-f009:**
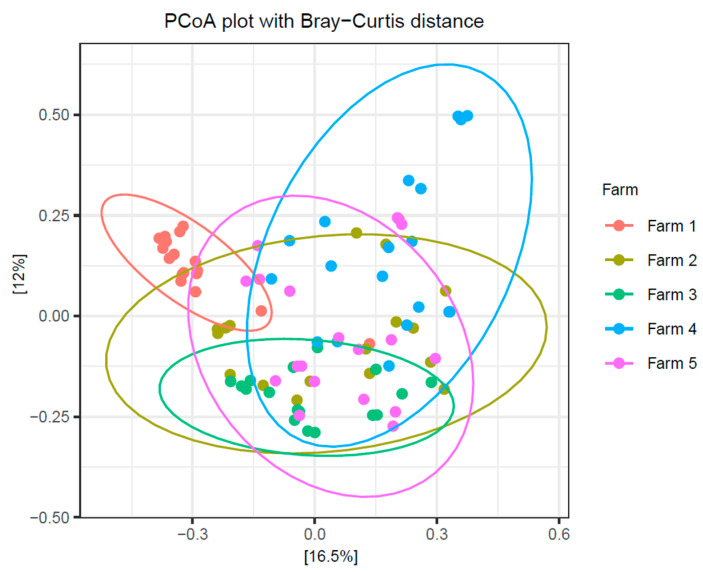
β-diversity analysis demonstrated discrete microbial clustering at the farm level. Microbial diversity was significant between farms (*p* < 0.05), with the strongest difference observed for Farm C (*p* < 0.01).

**Figure 10 antibiotics-14-01217-f010:**
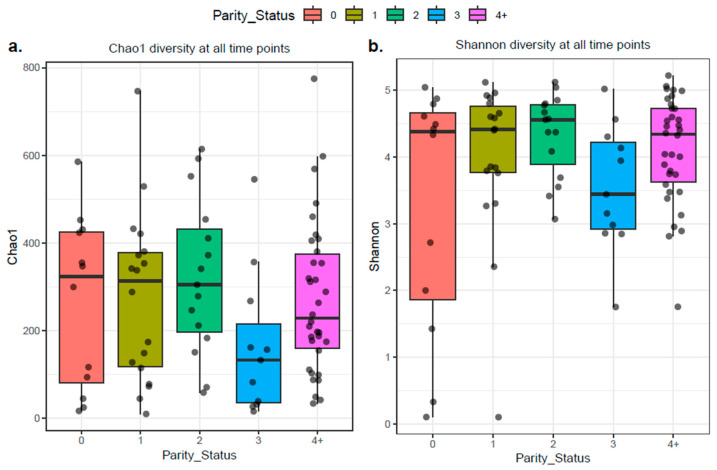
Colostrum samples grouped according to parity status (0, 1, 2, 3, 4+) showed no significant differences in alpha diversity. Both (**a**) Chao1 (estimated richness) and (**b**) Shannon (richness and evenness) indices were not significantly different across parity groups based on the Wilcoxon test. The *p*.adj values < 0.05 were considered significant.

**Figure 11 antibiotics-14-01217-f011:**
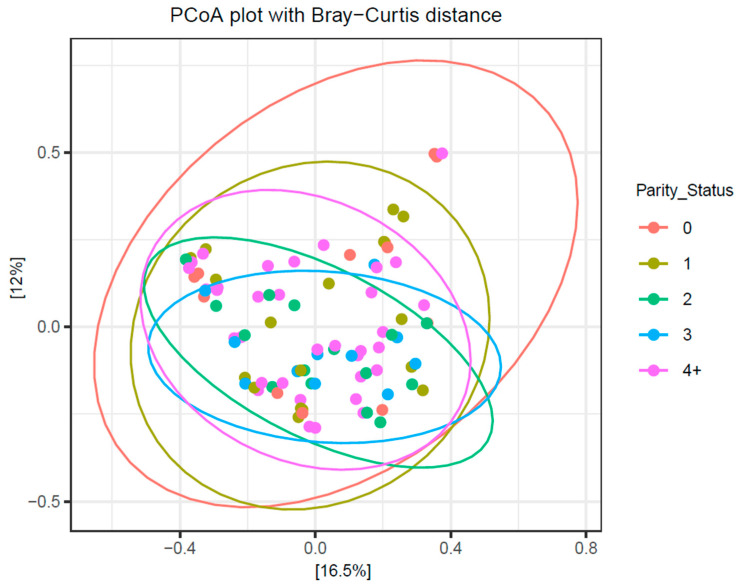
No significance in β-diversity was observed among colostrum samples when grouped by parity (*p* > 0.05).

**Table 1 antibiotics-14-01217-t001:** Significant associations among the ten most abundant genera in colostrum. Differential abundance analysis was conducted using MaAslin2, comparing antibiotic-treated groups (UBRORED or CEF) with the NOAB group as reference. Reported values include the model coefficient (β), standard error (SE), *p*-value, FDR adjusted q-value, total sample size (N), and the number of samples with non-zero abundance (N.not.0).

Feature	Value	Coef (β)	SE	*p*-Value	q-Value	N	N.Not.0
*Corynebacterium*	Ubro Red	1.327	0.494	0.0086	0.0251	90	89
*Facklamia*	Ubro Red	1.360	0.566	0.0184	0.0462	90	86
*Jeotgaliococcus*	Ubro Red	1.580	0.584	0.0082	0.0241	90	86
*Leuconostoc*	Cefquinome	3.766	1.231	0.0029	0.0099	90	43
*Staphylococcus*	Cefquinome	3.002	0.857	0.0007	0.0030	90	72
*Staphylococcus*	Ubro Red	2.546	0.851	0.0036	0.0115	90	72

## Data Availability

The raw data supporting the conclusions of this article will be made available by the authors on request.

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
