# Peer review of "The Microbial Composition of Bovine Colostrum as Influenced by Antibiotic Treatment"

_antibiotics, 2025, doi:10.3390/antibiotics14121217_

Round 1
Reviewer 1 Report
Comments and Suggestions for Authors
Following requests are necessary to be addressed to improve quality of the study.
Title:
- It would be better to remove “Irish” from the title, as the change in microbial composition is not specific to this breed of cow, and there is no supporting evidence for such an association. However, the type of cow can be mentioned in the Methods section.
Abstract:
- Line 9: The background sentence is overly concise and lacks sufficient detail; it should be expanded to provide more context and relevant information.
- Line 12: It is not clear how these five farms were selected were they already using this type of antibiotic, or was it introduced as part of the study? This point needs to be clearly explained in this section.
- Line 15: The sequence of events described in this section is not clearly presented. It should be stated that the antibiotic treatment was applied before calving, during the dry period, and that colostrum samples were subsequently collected after calving for microbial composition analysis.
- Line 16: Quantitative assessment of the microbial community is essential. It is unclear why the study relies only on qualitative 16S rRNA analysis rather than incorporating a more robust quantitative approach, such as Flow-FISH or other advanced techniques.
- Line 17: This sentence needs to be connected to the previous one or rewritten more clearly to improve understanding.
- Line 18: It is inappropriate to conclude that parity has no effect on microbial composition. The small sample size and the lack of detailed parity information are the main limitations in supporting this conclusion.
- Line 19: The significance of differences in the microbial composition of colostrum between the two groups of cows receiving different antibiotic treatments is unclear and requires additional clarification.
- Line 28: Based on the results, the study shows that withholding antibiotics during the dry period results in a distinct microbial composition. Consequently, the rationale for concluding that the study supports non-antibiotic alternative for dry-off is unclear. The specific type of alternative endorsed by the study should be clearly explained.
- A clear conclusion of the study needs to be presented.
- This section should address the study’s limitations and highlight areas requiring further investigation.
Introduction
In this section need to include:
- The composition of the colostrum microbial community.
-Identification of antibiotics that are safe for the microbiota versus those that have detrimental effects on these bacteria.
-Supportive evidence demonstrating that parity does not significantly influence microbial composition.
Results :
- Line 107, The results across the three groups need to clearly demonstrate the differences in microbial community composition.
- Line 157: Does this unexpected result have any interpretation or explanation?
- Line 229: Could you clarify the difference between the results presented in sections 2.1 and 2.3?
Methods:
- Line 602: It is unclear why there is a gap of about five years between the completion of the experiment and the submission of this draft for publication.
- Line 607 : Why was the study conducted across different farms? In your opinion, would it have been better to perform the study on a single farm and divide the cows into three groups according to the study design? This needs further explanation.
Discussion
- In need to explain if the result gotten is logical or some of them is unexpected
- Clearly explain the limitations of the study and recommend further research
- Include a section that highlights and explains the clear contributions and implications of this study.
Author Response
Title:
- It would be better to remove “Irish” from the title, as the change in microbial composition is not specific to this breed of cow, and there is no supporting evidence for such an association. However, the type of cow can be mentioned in the Methods section.
- Thank you for your suggestion, “Irish” has now been removed from the title and the breed of cows used in the study, Holstein Friesian is noted in the methods section.
Abstract:
- Line 9: The background sentence is overly concise and lacks sufficient detail; it should be expanded to provide more context and relevant information.
- Thank you for your comment the following has now been included on line 9-13. “Background/Objectives: Bovine colostrum is the initial milk produced by cows postpartum and contains an array of key nutritional, immune and microbial components that supports the calf’s physiological development, immune maturation and intestinal colonization. The composition and quality of colostrum can be influenced by multiple factors, including seasonal variation, breed, parity and farm management practices”
- Line 12: It is not clear how these five farms were selected were they already using this type of antibiotic, or was it introduced as part of the study? This point needs to be clearly explained in this section.
- Thank you for your suggestion. The farms selected were already following the described DCT strategies. No changes to the usual treatment of cows during the dry period were made as a result of our study. This is now clarified in the abstract line 15-17: “Bovine colostrum samples were collected from five Irish dairy farms that implemented different methods of dry cow therapy (DCT): natural or blanket.” and methods line 666-668: Farms enrolled in this study on a voluntary basis and no changes in the existing farm management practices were made due to participation in this study.
- Line 15: The sequence of events described in this section is not clearly presented. It should be stated that the antibiotic treatment was applied before calving, during the dry period, and that colostrum samples were subsequently collected after calving for microbial composition analysis.
- Thank you for your suggestion. We have now referenced the sequence of events in more detail in the abstract (line 17-22) and methods (line 610).
- Line 16: Quantitative assessment of the microbial community is essential. It is unclear why the study relies only on qualitative 16S rRNA analysis rather than incorporating a more robust quantitative approach, such as Flow-FISH or other advanced techniques.
- Thank you for your comment. We agree that quantitative assessment of the microbial community can provide valuable additional context. In this study, we used 16S rRNA sequencing as its widely applied and an appropriate method for characterising community composition and diversity that aligns with previous studies on the bovine colostrum microbiota, enabling a direct comparison. Our primary objective was to describe the microbial profile of colostrum and shifts as a result of DCT, rather than to quantify absolute microbial loads. We agree that Flow-FISH or other flow cytometry-based approaches would offer a complementary quantitative insight, but this method was not included in our original study design. We have now noted the value of these analyses for future studies in our discussion (line 637-639). “. In addition, quantitative techniques such as flow-cytometry based methods, including Flow-FISH, could complement sequencing data by providing direct measurements of absolute bacterial loads in colostrum”.
- Line 17: This sentence needs to be connected to the previous one or rewritten more clearly to improve understanding.
- Thank you for your suggestion. We have now updated line 23-26: “16S rRNA sequence analysis revealed Firmicutes, Actinobacteriota, Bacteroidota and Proteobacteria as the most abundant phyla across all treatment groups, with Acinetobacter, Corynebacterium, Facklamia, Jeotgalicoccus, Lactococcus, Leuconostoc, Psychrobacter and Staphylococcus dominating at a genus level.”
- Line 18: It is inappropriate to conclude that parity has no effect on microbial composition. The small sample size and the lack of detailed parity information are the main limitations in supporting this conclusion.
- Thank you for your comment. We have clarified that this finding applies within the context of this study. Line 26-27: “Parity did not significantly affect the microbial composition in this study, but antibiotic treatment did, with significant differences in microbial composition and diversity observed between cows treated with and without antibiotics”. Our analysis included 90 cows distributed across defined parity categories (0, 1, 2, 3, 4+), which provides sufficient representation to evaluate parity-associated differences at the community level. Within, this dataset parity did not show a significant effect. This sample size is in line with previous studies investigating parity and microbial composition (Lima et al, 2017., Zhu et al, 2023., Van Hese et al,, 2022.).
- Line 19: The significance of differences in the microbial composition of colostrum between the two groups of cows receiving different antibiotic treatments is unclear and requires additional clarification.
- Thank you for your comment the following changes have now been made on Line 27-31: “Cows receiving no antibiotics showed distinct microbial clustering compared with antibiotic-treated cows (B-diversity, p <0.001). Microbial diversity also differed between the antibiotic treated groups, with significant changes in both a-diversity (p < 0.01) and B-diversity (p < 0.001), suggesting choice of antibiotic may also influence the microbiota.”
- Line 28: Based on the results, the study shows that withholding antibiotics during the dry period results in a distinct microbial composition. Consequently, the rationale for concluding that the study supports non-antibiotic alternative for dry-off is unclear. The specific type of alternative endorsed by the study should be clearly explained.
- Thank you for your comment. We agree that our data does not directly support a specific non-antibiotic alternative for drying off. We have therefore removed the statement “The data support the use of non-antibiotic alternatives for drying off in cows” and instead refer to this as an area for future investigation” (line 40).
- A clear conclusion of the study needs to be presented.
- Thank you for your comment. The abstract has been restructured from line 34-43 to highlight a clear conclusion section.
- This section should address the study’s limitations and highlight areas requiring further investigation.
- Thank you for your suggestion, we have now included a brief statement in the abstract outlining key areas for future research line 38-40. “Future studies should elucidate strain level changes in the colostrum microbiota following on-farm antibiotic use, assess the associated risks of antimicrobial resistance, and explore non-antibiotic alternatives for drying off cows”. Consistent with the journal guidelines, the study’s limitations are addressed in more detail in the discussion section (lines 626-651).
Introduction
In this section need to include:
- The composition of the colostrum microbial community.
- Thank you for your comment. The composition of the colostrum microbial community is described in the introduction. Specifically, the dominant phyla and commonly reported genera are outlined in line 79-82, and the variability observed at a species level, including examples is detailed in lines 83-90.
-Identification of antibiotics that are safe for the microbiota versus those that have detrimental effects on these bacteria.
- Thank you for your comment. Identifying antibiotics that are “safe” for the microbiota is beyond the scope of this study and cannot be addressed based on our data. As far as we are aware, there is no established evidence or consensus defining specific antibiotics as microbiota-safe, as antibiotic choice and effects depend on multiple factors including spectrum of activity, dose, route, timing, host and target. Veterinary antibiotic safety assessments focus on animal health rather than microbiota preservation or AMR risk. We therefore do not include this statement in the introduction, but we have discussed broader antimicrobial use considerations and risks on lines 131-133 and 139-145.
-Supportive evidence demonstrating that parity does not significantly influence microbial composition.
- Thank you for your comment. The introduction has now been expanded on to include evidence on the effect of parity on microbial composition (line 114-120). “A multitude of factors can influence the composition and quality of bovine colostrum, including seasonal variation [6,37,38], maternal nutrition [14,39], breed [40,41] and herd size [42]. In addition, parity is known to influence the nutritional and immunological quality of colostrum [6,14,37,38,40,41,43,44]. However, evidence regarding its impact on the colostrum microbiota remains inconsistent [17,45,46]. Furthermore, although some influential factors are predetermined, others can be modified with farm management practices [38], in particular, DCT [38,43]”.
Results :
- Line 107, The results across the three groups need to clearly demonstrate the differences in microbial community composition.
- Thank you for your comment. The line 107 corresponds to the introduction. The differences in microbial community composition across the three treatment groups (NOAB, CEF, UBRORED) are presented in section 2.1 and 2.2. To further clarify these findings, we have added a brief summary statement of the major group changes on line 178-183. “These results demonstrate that colostrum from cows receiving no antibiotics during the dry period (NOAB) display a distinct microbial composition compared with both antibiotic-treated groups, characterized by lower Firmicutes and higher Bacteroidota. In addition, CEF and UBRORED colostrum also differ from each other across multiple taxa, including higher Bacteroidota in the UBRORED group, indicating that choice of DCT strategy can influence the microbial community of colostrum.”
- Line 157: Does this unexpected result have any interpretation or explanation?
- Thank you for your comment. We have now expanded on this result in the discussion to contextualise the lower abundance of Firmicutes in the NOAB group, including potential AB effect and environmental factors. Line 343-347: “The lower abundance of Firmicutes in the NOAB group is unexpected and the cause remains unclear. However, this may be a result of a more balanced community profile, allowing competing Bacteroidota to occupy a greater proportion of the colostrum microbiota. Additionally, farm level factors may contribute to the observed pattern as Firmicutes are known to vary with housing, hygiene and farm management conditions”.
- Line 229: Could you clarify the difference between the results presented in sections 2.1 and 2.3?
- Thank you for your question. Section 2.1 and 2.3 report different aspects of the microbial community. Section 2.1 presents the composition of colostrum microbiota, including the taxonomic profiles and differences in relative abundance of specific phyla. Section 2.3 outlines the diversity of these communities, assessing the intra-sample richness/evenness (alpha diversity) and inter-sample dissimilarity (beta diversity). More simply, section 2.1 tells us what microbes are present (composition), and section 2.3 tells us how diverse/different they are (diversity).
Methods:
- Line 602: It is unclear why there is a gap of about five years between the completion of the experiment and the submission of this draft for publication.
- Thank you for your question. The samples were collected, processed and sequenced within a 12-month period. The subsequent analysis and manuscript preparation were completed later due to unavoidable delays unrelated to the experimental work. These delays did not affect sample integrity, sequencing quality or the reliability of the results.
- Line 607 : Why was the study conducted across different farms? In your opinion, would it have been better to perform the study on a single farm and divide the cows into three groups according to the study design? This needs further explanation.
- Thank you for your question. This study was conducted across different farms because each farm was already implementing a distinct DCT protocol as part of their routine management. This allowed us to investigate the effects of these practices without altering farm procedure or introducing experimental interventions. Conducting the study on a single farm and allocating cows to different DCT protocols would have required changes to the standard mastitis control practices used by these farmers, which was not feasible and could have resulted in their withdrawal from the study or raised ethical and health concerns. Using multiple farms therefore enabled us to examine the impact of DCT under typical commercial conditions whilst maintaining ongoing farm management practices.
Discussion
- In need to explain if the result gotten is logical or some of them is unexpected
- Thank you for your comment. We have now further clarified in the discussion which findings are expected based on previous studies and which findings are unexpected. These additions have been incorporated on line 330-331 (phylum level abundances), line 343: the lower abundance of Firmicutes in NOAB, the expected disparity in genus level distribution (line 365), the increased abundance of Staphylococcus in AB treated groups (line 409) and Leuconostoc in CEF-group (line 456).
- Clearly explain the limitations of the study and recommend further research
- Thank you for your suggestion, the limitations of the study and recommendations for future research is now outlined on line 626-648. “A key limitation in our study is the use of 16S rRNA sequencing at A key limitation in this study is the use of 16S rRNA sequencing at a single time point. Future studies should incorporate longitudinal sampling and higher resolution sequencing approaches. Shotgun metagenomics or whole-genome sequencing (WGS) would provide more detailed insights into the influence of DCT, strengthen the data on on-farm antibiotic usage, provide greater insight on the potential resistome of bovine colostrum and enable a greater understanding of species level variability and functionality. Identifying the source of ARGs on dairy farm is critical to mitigate the global spread of antibiotic resistance. Further investigations is needed to determine the origin and mechanism of transmission, including horizontal and vertical gene transfer. To this end, the inclusion of both environmental (bedding, water, soil) and faecal samples would provide a more comprehensive insight into resistome dynamics on farm. In addition, quantitative techniques such as flow-cytometry based methods, including Flow-FISH, could complement sequencing data by providing direct measurements of absolute bacterial loads in colostrum. Furthermore, the inclusion of an additional treatment group of cows with a positive infection status would indicate how infection modulates microbial responses to drying off. Future studies should monitor SCC and IgG levels, to confirm udder health and evaluate any secondary effect of DCT on colostrum quality. IgG testing can be performed on-farm using a Brix refractometer, however for improved accuracy, laboratory-based methods such as radial immunodiffusion assay or ELISA can be performed [134,135]. For SCC testing, several on farm techniques are available, including the California Mastitis Test, the Porta SCC test and the Delaval Cell Counter [136]. In addition, an automated fluorescence-based cell counter, the Fossomatic, is widely used and provides high-throughput, accurate results”.
- Include a section that highlights and explains the clear contributions and implications of this study.
- Thank you for your comment. We have now added a paragraph to the end of the Discussion, separate from the Conclusion, that outlines the key contributions of our study and their implications for dying off. Line 651- 658 “This study provides new insight into how different DCT strategies influence the microbial composition of bovine colostrum under commercial dairy farm conditions. Our findings demonstrate that natural drying off is associated with a distinct colostrum microbiota, without an increased abundance of mastitis-associated genera. These results have practical implications for farm management practices, supporting the current move towards reduced prophylactic antibiotic administration and highlighting the need for further development of targeted non-antibiotic alternatives to maintain udder health”
Reviewer 2 Report
Comments and Suggestions for Authors
The procedures for sanitising the teat during colostrum collection should be detailed, as the results may be affected by contamination.
How was it determined that the cows were actually healthy?
A more detailed presentation of the data would provide a more comprehensive and functional assessment of colostrum quality, in addition to the microbial profile already established.
Expand the discussion, and, if possible, supplement the analysis to include the topic of Antimicrobial Resistance.
Author Response
The procedures for sanitising the teat during colostrum collection should be detailed, as the results may be affected by contamination.
- Thank you for your comment. The procedures used to minimise the risk of contamination during colostrum collection are detailed in Section 4.2. These include pre-dipping with iodine, drying the teats and scrubbing with wipes soaked in ethanol prior to sterile sample collection, following the recommendation of the National Mastitis Council. Line 688-699: “All farmers were provided with and followed instructions for sterile sample collection, in accordance with the recommendations from the National Mastitis Council’s Laboratory Handbook on bovine mastitis [137]. In brief, colostrum samples were collected at the first milking within 1 h of calving. To ensure mammary gland stimulation, the first streams of colostrum from each mammary quarter were discarded (fore-stripping). Pre-dipping was then carried out by dipping the teats in iodine tincture. Teats were dried and scrubbed with wipes soaked in 70 % alcohol. Finally, 15 mL of colostrum were collected in sterile falcon tubes (Starstedt). Each sample was labelled with the cow’s identification number and the date of calving”.
How was it determined that the cows were actually healthy?
- Thank you for your question. Cows were visually inspected by quality-assurance certified members of the Irish Food Board (Bord Bia) throughout the dry period, calving period and immediately prior to colostrum collection and no clinical signs of infection were observed. We note that SCC measurements would have provided additional objective indicator of udder health, however SCC testing was not included in the original study design. We have acknowledged this limitation on 642-643.
A more detailed presentation of the data would provide a more comprehensive and functional assessment of colostrum quality, in addition to the microbial profile already established.
- Thank you for your suggestion. Compositional measures such as IgG, lactoferrin, macronutrients content, or TCC/TPC/SCC were not included in our original study design. We agree that these parameters would complement the microbial analysis performed here. We have acknowledged this limitation in the discussion and noted that future studies should incorporate SCC and IgG measurements to provide a more comprehensive assessment of colostrum quality on line 642-649. “Future studies should monitor SCC and IgG levels, to confirm udder health and evaluate any secondary effect of DCT on colostrum quality. IgG testing can be performed on-farm using a Brix refractometer, however for improved accuracy, laboratory-based methods such as radial immunodiffusion assay or ELISA can be performed [134,135]. For SCC testing, several on farm techniques are available, including the California Mastitis Test, the Porta SCC test and the Delaval Cell Counter [136]. In addition, an automated fluorescence-based cell counter, the Fossomatic, is widely used and provides high-throughput, accurate results.”
Expand the discussion, and, if possible, supplement the analysis to include the topic of Antimicrobial Resistance.
- Thank you for your suggestion. We agree that antimicrobial resistance is an important consideration in the context of our study. Due to sequencing depth limits and the use of 16S sequencing, we were unable to assess antimicrobial resistant genes directly. We have now expanded on our Discussion to address the relevance of AMR in DCT and to highlight this as a limitation of our study. We also recommend that future studies incorporate shotgun metagenomics and multiple sample sites to characterise resistome dynamics on the farm line 629-637. “Shotgun metagenomics or whole-genome sequencing (WGS) would provide more detailed insights into the influence of DCT, strengthen the data on on-farm antibiotic usage, provide greater insight on the potential resistome of bovine colostrum and enable a greater understanding of species level variability and functionality. Identifying the source of ARGs on dairy farm is critical to mitigate the global spread of antibiotic resistance. Further investigations is needed to determine the origin and mechanism of transmission, including horizontal and vertical gene transfer. To this end, the inclusion of both environmental (bedding, water, soil) and faecal samples would provide a more comprehensive insight into resistome dynamics on farm”.
Reviewer 3 Report
Comments and Suggestions for Authors
The research are designed good by authors. There are few mistakes
- Why do author prefer to perform 16sRNA sequencing? Because this method is not good in the findings differences at the species or strain level.
- The legends in some figures are repeated, it will be better to carefully look through figures size and legends.
- it is suggested to carefully review the numbering and style of the manuscript according to the journal guidelines.
Author Response
The research are designed good by authors. There are few mistakes
- Why do author prefer to perform 16sRNA sequencing? Because this method is not good in the findings differences at the species or strain level.
- Thank you for your question. We selected16s rRNA sequencing because it is a well-established approach for characterizing overall microbial community structure and broad taxonomic patterns across large sample sets, and it is consistent with the methodologies used in previous colostrum microbiome studies. Our objective was to describe community level differences associated with DCT rather than to resolve strain level variations. We agree that shotgun metagenomics would provide higher taxonomic and functional resolution, and we have added a statement in the discussion to highlight this limitation and recommend deeper sequencing approaches for future studies. Line 626-632: “A key limitation in our study is the use of 16S rRNA sequencing at A key limitation in this study is the use of 16S rRNA sequencing at a single time point. Future studies should incorporate longitudinal sampling and higher resolution sequencing approaches. Shotgun metagenomics or whole-genome sequencing (WGS) would provide more detailed insights into the influence of DCT, strengthen the data on on-farm antibiotic usage, provide greater insight on the potential resistome of bovine colostrum and enable a greater understanding of species level variability and functionality. “
- The legends in some figures are repeated, it will be better to carefully look through figures size and legends.
- Thank you for your comment. The caption of figures have now been amended across the manuscript and legends reviewed to ensure they accurately align with the figures.
- it is suggested to carefully review the numbering and style of the manuscript according to the journal guidelines.
- Thank you for your suggestion. The manuscript has been reviewed, and the abstract structure has been updated to reflect the journal guidelines (line 9-43). The remaining sections have been reviewed to ensure numbering and style is in line with journal requirements.
Round 2
Reviewer 1 Report
Comments and Suggestions for Authors
Most of the requested corrections were performed. Thank you